# Organization and Management of Conservation Programs and Research in Domestic Animal Genetic Resources

**Juan Vicente Delgado Bermejo [1]**, **María Amparo Martínez Martínez [2]**, **Guadalupe Rodríguez Galván [3]**, **Angélika Stemmer [4]**, **Francisco Javier Navas González [1,\*]** and **María Esperanza Camacho Vallejo [5]**

[1] Department of Genetics, Faculty of Veterinary Sciences, RBI/CONBIAND, University of Córdoba, 14014 Córdoba, Spain; juanviagr218@gmail.com
[2] Animal Breeding Consulting, S.L., Córdoba Science and Technology Park Rabanales 21, 14014 Córdoba, Spain; amparomartinezuco@gmail.com
[3] Institute of Indigenous Studies, Autonomous University of Chiapas, 29264 San Cristóbal de las Casas, Chiapas, Mexico; gr.galvan2010@hotmail.com
[4] Faculty of Agricultural Science and Livestock, San Simon Major University, Cochabamba, Bolivia; gelistemmer@hotmail.com
[5] Instituto de Investigación y Formación Agraria y Pesquera (IFAPA), Alameda del Obispo, 14005 Córdoba, Spain; mariae.camacho@juntadeandalucia.es
\* Correspondence: fjng87@hotmail.com; Tel.: +34-651-679-262

**Abstract:** Creating national committees for domestic animal genetic resources within genetic resource national commissions is recommended to organize in situ and ex situ conservation initiatives. In situ conservation is a high priority because it retains traditional zootechnical contexts and locations to ensure the long-term survival of breeds. In situ actions can be based on subsidies, technical support, structure creation, or trademark definition. Provisional or permanent relocation of breeds may prevent immediate extinction when catastrophes, epizootics, or social conflicts compromise in situ conservation. Ex situ in vivo (animal preservation in rescue or quarantine centers) and in vitro methods (germplasm, tissues/cells, DNA/genes storage) are also potential options. Alert systems must detect emergencies and summon the national committee to implement appropriate procedures. Ex situ coordinated centers must be prepared to permanently or provisionally receive extremely endangered collections. National germplasm banks must maintain sufficient samples of national breeds (duplicated) in their collections to restore extinct populations at levels that guarantee the survival of biodiversity. A conservation management survey, describing national and international governmental and non-governmental structures, was developed. Conservation research initiatives for international domestic animal genetic resources from consortia centralize the efforts of studies on molecular, genomic or geo-evolutionary breed characterization, breed distinction, and functional gene identification. Several consortia also consider ex situ conservation relying on socioeconomic or cultural aspects. The CONBIAND network (Conservation for the Biodiversity of Local Domestic Animals for Sustainable Rural Development) exemplifies conservation efficiency maximization in a low-funding setting, integrating several Latin American consortia with international cooperation where limited human, material, and economic resources are available.

**Keywords:** local breeds; management; in situ conservation; rescue centers; germplasm banks; research

## 1. Introduction

In the 20th century, the loss of domestic animal diversity became dramatic, and sensitivity to conservation issues for the domestic world started to be borrowed from wildlife conservationists. However, conservation efforts in domestic animals were delayed until the Food and Agriculture Organization (FAO) mentioned its interest in the matter in 1948 [1] as a result of the rising awareness of the loss of breeds adapted to harsh conditions. At this time, the discussion about the preservation of domestic animal breeds started growing, including both ex situ and in situ approaches. It would not be until 1974 [2], during the First World Congress on Genetics Applied to Livestock Production, when the general idea of breed conservation was consolidated. Prior to this event, the general idea of conservation was raised for discussion in chickens around 10 years before [3–5].

The need for the organization and coordination of conservation processes became clear at two levels. From the governmental point of view, the FAO assumed a key role [6], while, from the non-governmental point of view, the British Rare Breed Survival Trust (RBST), founded in 1973, became a pioneer in domestic animal conservation programs [7].

Global coordination was not reached until 1990, when the FAO's council recommended the development of a global program for the sustainable management of animal genetic resources. General coordination became more concrete during the International Technical Conference on Animal Genetic Resources in Interlaken, Switzerland in 2007 [8]. After this conference, the Global Plan of Action for Animal Genetic Resources was adopted [9]. The real inflection point for general sensitivity in favor of domestic animal genetic resources with food and agricultural interest was the Convention on Biological Biodiversity [10] in 1992 in Rio de Janeiro under the umbrella of the United Nations Conference on Environment and Development.

Its main recommendations were the conservation of biodiversity, ensuring its sustainable use, and the fair and equitable sharing of the benefits arising from its utilization, including agricultural biodiversity. Further recommendations regarding the access to genetic resources were presented in the Nagoya Protocol in 2014 [11]. The contents of this protocol would be crucial to mediate national and international regulations on breeding and domestic animal genetic resources [12].

At the non-governmental level, the aggregation of national and regional organizations began under the umbrella of Rare Breed International (RBI) beginning in 1990. This organization was recently transformed into a representative organization, acting as a platform for regional and national organizations [13].

Several actions are now developed to organize and coordinate conservation activities at all levels: designing strategies, proposing in situ and ex situ actions, and promoting international research. In the present overview, a description of the main aspects of these actions is presented.

## 2. Organization of the Conservation of Domestic Animal Genetic Resources

Several important reasons justify the effective management of agrobiodiversity by governments and non-governmental organizations [14]. The first of these reasons stems from the fact that domestic animal genetic resources supply high-level nutrients and essential industrial products. They are sources of wealth and the base of sustainable development in countries as a guarantee of adaptation to eventual future changes such as those related to climatic conditions [15]. In many countries, local conservation actions are usually isolated and scarcely coordinated; thus, the limited resources that are available for conservation are not rationally spent. For this reason, a national conservation program involving plants, microorganisms, and animals is necessary.

Agrobiodiversity plays an important role in national food security and sovereignty, particularly in difficult territories, but also contributes an important part of the cultural and natural patrimony of countries. Agrobiodiversity is in the hands of two main stakeholders. One of these is agribusiness, whose purpose is the development of markets by using highly specialized but low-biodiversity populations. The second is traditional production systems that pursue the qualitative activation of local markets stimulating local consumption and the creation of protected trademarks using highly

diverse populations adapted to different conditions (Figure 1). Agribusiness is responsible for food security while traditional production contributes to food sovereignty [16]. Food sovereignty adds to the concept of food security that food is more than just a commodity, but it is linked to traditional knowledge, respecting local systems in equilibrium with nature [17].

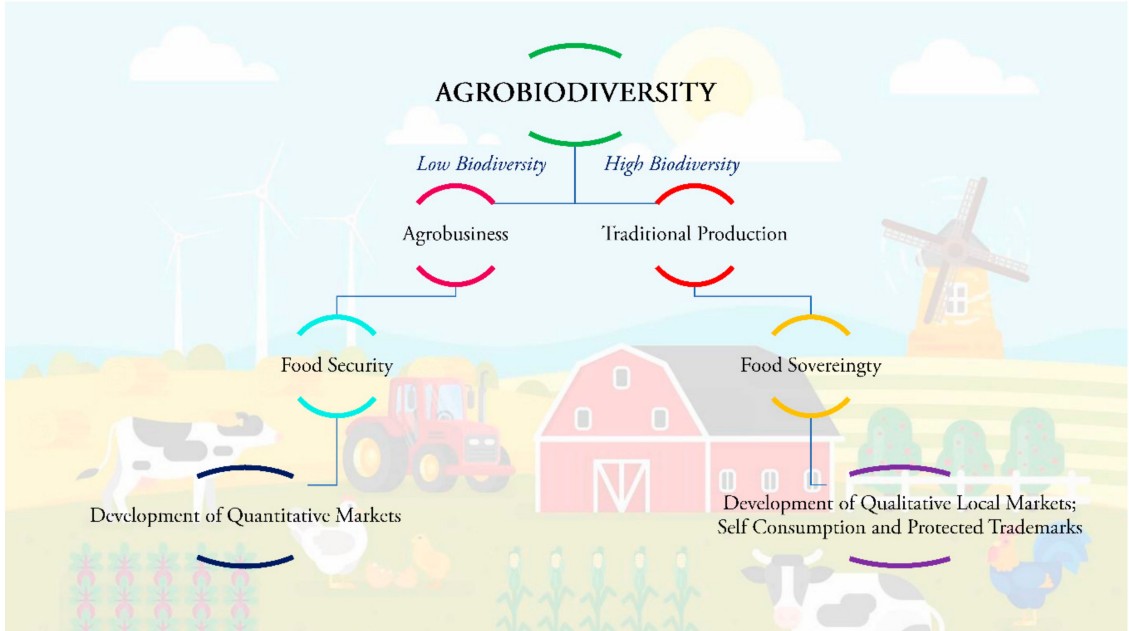

**Figure 1.** Distribution of agrobiodiversity.

National commissions on biodiversity must involve three specialized committees: plants, microorganisms, and animals. The national committee of domestic animal biodiversity would be responsible for matters related to domestic animal genetic resources conservation, coordinating all the national activities under development in the country in question (Figure 2).

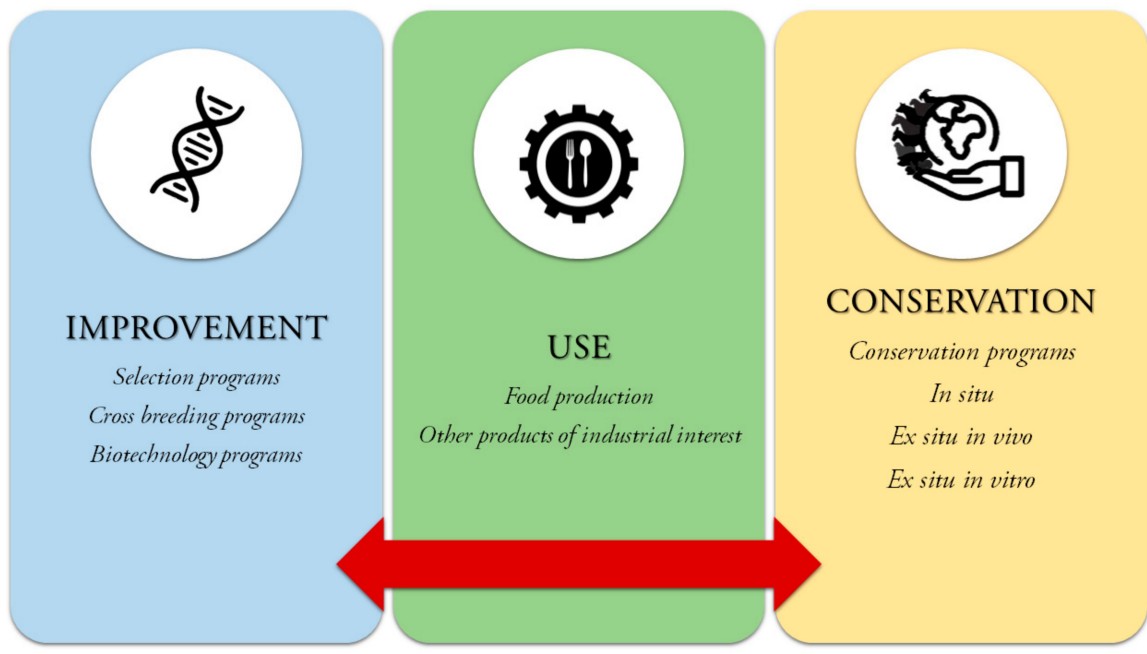

**Figure 2.** Subjects of actions of the national committee of domestic animal biodiversity.

Breeder associations, cooperatives, clubs, Non-Governmental Organizations (NGOs), universities and research centers, local administrations, and sometimes individual farmers are the primary units to run projects and programs based on in situ and/or ex situ methods [18]. The objective is the maintenance of domestic animal diversity through the conservation of breeds and other genetic structures and/or maximizing intrapopulation genetic variability by means of the implementation of correct genetic management. Their coordination is the mission of the national committee of domestic animal biodiversity.

The actions aimed at conserving domestic animal diversity must be planned and coordinated in the context of conservation programs that target specific populations (breeds, varieties, ecotypes) [8]. Individual programs must be integrated into regional programs at a territorial level, which must optimize the available resources. All territorial programs must be coordinated by the national committee of domestic animal biodiversity as the main entity responsible for the national program of domestic animal conservation, using rules and regulations. National programs must be integrated for the execution of general programs, information and experience exchange, and the development of general infrastructure such as databases, technological rules, and others [8].

The national committee of domestic animal biodiversity must comprise three aspects, the first is administrative, and involves representatives from the ministry of agriculture and other ministries and territorial representatives, the second involves representatives from farmer and breeder associations, cooperatives, NGOs, and industry, and the third involves scientific researchers with experience and expertise. The targets of this committee are complex but can be summarized as follows:

1. Management of official recognition of breeds and breeder associations responsible for herd books and breeding programs.
2. Development, management, and updating of the national inventory and databases on breeds and breeder associations.
3. Establishment and maintenance of the national alert system.
4. Management of the national germplasm bank and coordination of the territorial banks.
5. Establishment of the national network of ex situ in vivo conservation centers.

The official recognition of both breeds and breeder associations must be by defined protocols that are published and transparent. Most countries lack defined criteria to implement these important efforts, and the decisions of recognition can be arbitrary and unsupported by scientific and technical criteria.

Breed recognition must start with the development and presentation of a report to the committee which includes a historical collection of proof (photographs, paintings, texts, others). The most important part of the report concerns the breed's characterizations, developed by a specialized research group and including genetic (microsatellites, single nucleotide polymorphisms (SNPs)), morphological (zoometry, breed pattern, others) [19], and functional (growth, meat, milk, behavior, sports, others) characteristics [20]. These characterize the breed's differentiation from other populations. Proposed rules for the herd book and breeding program management must be included.

The official process for recognition of breeder associations demands a list of members, a registry of animals belonging to the breed and managed by the association, a list of technical and human resources available to develop and manage the herd book, and a breeding program.

The committee will study the proposals in the report, and it will approve if all necessary requirements are met. The committee classifies the populations according to their conservation status (extinct, critical, critical maintained, endangered, endangered maintained, not at risk, and unknown), following the FAO's recommendations [21].

Once the breed and the breeder associations re officially recognized, governments must develop official national registries, which must be updated periodically. These registries must be the legal base to receive technical support and subsidies.

The official national register of breeds is the essential inventory for the management of animal genetic resources within the country. Apart from the basic list of official breeds, governments must

develop an open-access database acting as an efficient platform to collect, organize, and disseminate information about breeds and to keep this information permanently updated. This platform must be compatible with other international databases such as the FAO's Domestic Animal Diversity Information System (DAD-IS) [22] or the European Farm Animal Biodiversity Information System (EFABIS).

The committee must use the information updated in the platform to monitor the breed's situation every year [23,24] by using conventional methods [25] or modern methodologies based on the multi-criteria analysis [26,27]. The committee must issue an annual national general report about the situation of national animal genetic resources, presented to the national commission and compiled together with those from microorganisms and plant committees to publish a joint annual national report on agrobiodiversity.

The national alert system is a structure of the committee, formed by the representatives of the territories in the national network. Its responsibility is the early detection of threats of extinction for individual breeds or collectives. This national alert can be motivated by general causes such as epizootics, natural disasters, and social conflicts, among others, but also by concrete causes affecting individual populations [28].

The national alert system operates as follows:

1.  When an alert is detected in the territories, the local official responsible collects information and immediately contacts the national coordinator.
2.  The national coordinator calls an extraordinary meeting of the committee.
3.  A general analysis of the alert is developed at this extraordinary meeting, and a plan of action is proposed.
4.  The plan of action is implemented.
5.  Progress of the actions is followed in successive general meetings of the committee.

Another responsibility of the committee is the organization and coordination of the national network of germplasm banks, a task that is difficult in small developing countries. The network comprises a central national germplasm bank and duplicated territorial germplasm banks. The distance among germplasm duplicate collections must be enough to ensure the conservation of at least one of them in an eventual catastrophe [29]. The committee must maintain an inventory of the stored resources at all germplasm banks and create a system for its permanent updating. The committee must prepare directives about recommended uniform methodologies for all species, with norms of identification, and storage and legal ownership of the collections.

A national inventory of potential ex situ in vivo centers must be created by the committee, including a list of private and public centers with the capacity to maintain live animals out of their traditional locations for a certain period of time. The aim of this action is to save them from extinction caused by drastic effects such as quick reduction of the census, epizootics and eradication campaigns, conflicts, or natural disasters [30].

## 3. International Management of the Conservation of Domestic Animal Genetic Resources

All the national initiatives need an international umbrella to serve as a platform for the exchange of information and experiences and to propose technical and methodological recommendations. This is especially relevant for the creation of infrastructures available at an international level, thus coordinating domestic animal genetic conservation at worldwide levels. These activities are developed in two frameworks: the governmental level and the non-governmental level.

### 3.1. Governmental Context

The roles comprising the governmental context are generally assumed by the Food and Agricultural Organization (FAO) of the United Nations. During the 1990s, FAO prepared the first global approach for the sustainable management of animal genetic resources (AnGR). In 1993, the FAO designed a global strategy with the aim to organize and coordinate the initiatives contributing to domestic animal

conservation and use [6]. The FAO published reference guidelines for the development of national farm animal genetic resource management plans, regarding the tools for genetic characterization of the populations [31] and the design of programs of conservation [32].

The FAO then implemented the global strategy, creating an international network of national and regional focal points. The FAO asked member countries to prepare country reports on the status and trends of their resources, the current and potential contributions of animals to food, agriculture, and rural development, and the status of national capacities to manage these resources. These reports were presented by the countries and were compiled by the FAO, publishing a general report which was presented as the State of the World's Animal Genetic Resources for Food and Agriculture [9].

The Interlaken Conference resulted in the adoption of the Global Plan of Action for Animal Genetic Resources and the Interlaken Declaration [8]. A global data bank was developed to store country information on national domestic animal genetic resources, and it was named the Domestic Animal Diversity Information System (DAD-IS). DAD-IS was developed as a software application to allow national focal points to add the data that would later be made publicly available. The system was recently updated (FAO, 2019).

The Global Plan of Action provides an international mandate to proceed with the characterization, inventory, monitoring, conservation, development, and sustainable use of domestic animal genetic resources, and to ensure that the benefits of their use and the responsibilities of their conservation are shared fairly and equitably.

In this context, the north–south transference of funds was recommended, as well as expertise and political support to implement the intended activities through bilateral, regional, and multilateral cooperation.

*3.2. Non-Governmental Context*

Although most non-governmental organizations are included or related to the FAO global program, a special survey on their role was carried outs in the present review. These NGOs play an important role in the development of conservation programs, especially in those countries where governmental activities are not efficient.

As reported by Ramsay et al. [33] from their experiences in South Africa, the role of NGOs and other types of private associations is very important for endangered populations, because these institutions can mobilize their own budgets very easily. They have a faster reaction when compared to governmental structures, as bureaucracy usually delays actions.

The British Rare Breed Survival Trust (RBST) [7], founded in 1993, was the pioneer organization to have domestic animal conservation programs among its priorities. After its foundation, the RBST experienced quick growth and an increase in its activities. In the context of the British cultural influence, other countries created and developed their own NGOs devoted to domestic animal genetic resource conservation. In the United States of America, the Livestock Conservancy (LC) [34] was founded in 1977 with the same purpose as the RBST. Something similar occurred in South Africa with the foundation of the Farm Animal Conservation Trust (FACT) [35]. Other examples of associations created under the scope of the British influence are the Rare Breed Conservation Society of New Zealand, founded in 1988 [36], Heritage Livestock Canada, founded in 1987 [37], and the Rare Breed Trust in Australia, founded in 1973 [38]. All these reached a high dissemination of the conservation concept in their countries, creating a large-scale network. They published a large number of books and other material. These organizations promoted specific conservation programs which saved hundreds of breeds from extinction.

Outside the British influence, other important NGOs appeared on the European continent. On the Iberian Peninsula, the Spanish Society for Domestic Animal Conservation (SERGA) was created in Spain in 1984 [39]. Some years later, the Portuguese Society for Domestic Animal Conservation [40] was founded as a sister NGO in Portugal. Today, both organizations cooperate in their activities. These organizations develop the national inventories of native breeds and promote the official

recognition of breeds, providing technical support to breeder associations. These NGOs organize an Iberian scientific event concerning all aspects of conservation every two years.

The most active NGO in continental Europe is the SAVE foundation (Safeguard for Agricultural Varieties in Europe), founded in 1993 to act as an umbrella NGO for the conservation of agrobiodiversity in Europe. SAVE foundation involves both plants and animals. This organization includes 25 national or local organizations seeking common purposes [41], all of them developing conservation programs using private funds, but also benefiting from European Union (EU) funds. The SAVE foundation was important in the organization of the European Livestock Breeds Ark and Resdcue Net (ELBARN) to study the ex situ in vivo conservation.

Another important umbrella organization is the CONBIAND Network [42], which was founded in 1990 as the XII-H Iberoamerican Science and Technology Program for Development (CYTED) Program Network for the characterization, conservation, and valuation of the Ibero-American domestic animal genetic resources and their traditional management systems. This NGO involves partners from 23 countries. CONBIAND stands out for its subsidiary networks (TRANSIBER, GASTRIBER, FORO GANADERO) and for its consortia on biodiversity studies in domestic animal species [42]. The organization supports several governments of the Latin America region in the implementation of their conservation national programs, establishing important processes of transference to the sector.

Perhaps the most important activity of CONBIAND network relies on its formative character. More than 300 specialists in conservation were trained, together with the development of more than 20 PhDs. Twelve formative books dealing with several Latin American domestic species were published.

To coordinate of all these national and regional organizations at a worldwide level, Rare Breeds International (RBI) was formed in 1989 [13]. RBI assisted in the implementation of the Global Plan of Action for Animal Genetic Resources with the FAO [14]. In 2018, RBI was transferred from the United Kingdom to Spain and started the implementation of new projects of the organization.

### 3.3. Conservation Programs

Among the methodologies of conservation, in situ methods imply the maintenance of animal populations in their own social and ecological context at a sufficient number to ensure enough levels of biodiversity to guarantee their survival. Oppositely, ex situ in vivo conservation is the relocation of endangered populations out of their social and ecological context to protect them under a temporary or permanent situation to prevent them from extinction. Ex situ in vitro conservation is the utilization of reproductive biotechnology to permanently conserve germplasm belonging to representative individuals of the populations with the aim of ensuring the maintenance of endangered populations or the restoration of such populations if all its living effectives disappear. All these methods must be coordinated and appropriately organized in conservation integral programs, including territorial, national, and international levels.

### 3.4. Organization of in Situ Conservation

The concept of a domestic breed is highly controversial, and several definitions can be found in the literature. A breed is a subspecies population formed by individuals showing heritable morphological, functional, and genetic characteristics which identify them with respect to other populations of the species in a sufficient way to be socioculturally and/or administratively recognized. That is, breeds are stores of genetic diversity; thus, when a breed becomes extinct, many original genes disappear. All our genetic patrimony is necessary for human survival and must be preserved. Substructures such as varieties, ecotypes, lines, and strains can also exist within breeds. All of them are intermediate steps of biodiversification.

Breeds are safe from extinction when they are completely integrated into their productive context [43]. They and their products must be recognized and valued by society. Breeds and their management systems must count on strong integration within the environment in which they coexist,

and they must count on the necessary structures for their successful breeding [44]. For this reason, in situ conservation is always an important first option.

The most relevant actions of in situ conservation are the following [32]:

1.　An in situ conservation program that maximizes genetic diversity, with accurate selection of breeding animals to provide the maximum variability possible to the population.
2.　The creation of livestock structures (breeder associations, phenotype recording systems, animal health, germplasm bank, others).
3.　Compensation for loss of income from breeding local breeds with lower productive potential.
4.　Financial resources for actions that are otherwise impossible due to the economic situation of small stakeholders.
5.　Measures to value the specific products derived from these breeds, with official definition of these products and the creation of short specific market chains and protected trademarks.
6.　Searches for other alternative uses.

Pascual and Perrings [45] proposed incentives as the most effective action for agrobiodiversity conservation. They suggested a different strategy from traditional incentives for production, which tends to only benefit intermediaries. Their proposal involves nascent market creation incentive mechanisms for biodiversity conservation, based on payments or rewards for ecosystem services, direct compensation payments, land-use development rights, and actions for biodiversity conservation. This philosophy was followed by Pattison et al. [46], who evaluated the cost of conservation for the implementation of a compensatory program in a population of pigs in México. The compensatory measures of conservation must be appropriately evaluated [47] and must be included in specific national regulations and rules under the scope of national conservation programs [48].

Serious differences exist between developed and developing countries in terms of the implementation of in situ conservation programs. The European Union policies of incentives for rare breed conservation are well known and continuously criticized [49]. In developing countries, initiatives are scarce [50], and only isolated programs are described.

### 3.5. Organization of Ex Situ Conservation

Sometimes, in vivo conservation is not possible or is not sufficient. In such cases, other technologies must be applied to ensure the long-term survival of populations that lost all of their living animals or whose populations present a low effective size that does not ensure the maintenance of sufficient internal genetic diversity to guarantee their long-term survival [9].

In these circumstances, two strategies can be implemented. One is the maintenance of live animals outside their traditional context (ex situ in vivo conservation) [32]. A second is the cryopreservation of germplasm (ex situ in vitro conservation) [29].

### 3.5.1. Ex Situ *In Vivo* Organization

Endangered populations usually present small and concentrated censuses. This predisposes them to suffering from sudden and intense external effects such as natural catastrophes, social changes, or any other action that produces a drastic population reduction (indiscriminate crossbreeding, massive slaughtering). In these populations, ex situ in vivo facilities become compulsory. These facilities present the advantage of allowing observation of living animals. These methods maintain animals at the margins of their traditional context, which facilitates their eventual reintroduction. This method counts on a network of centers with facilities to supply emergency places for important breeding animals and to support their breeding and public display.

There are three types of these centers, each with different specialization. Ark centers permanently maintain individuals belonging to populations which are practically extinct. Rescue centers are prepared to provisionally maintain populations to save them until they reach a status that enables

their reintroduction back to their contexts. Quarantine stations provide facilities to maintain isolated animals to prevent them from the contagion of epizootics.

The procedure of ex situ in vivo conservation was proposed by the ELBARN consortium [30] as follows:

1.  Individuals or groups of individuals belonging to a particular population are identified to be at imminent risk.
2.  People, technicians, or institutions raise the alarm to the territorial authorities; they issue this to the national entity, which calls an extraordinary meeting of the national committee.
3.  The situation is evaluated, and the rescue action is determined.
4.  Ark, rescue, or quarantine centers (depending on the profile of the alert) are identified.
5.  An agreement is signed among the responsible parties.

It is impossible to predict the reclusion time at ark, rescue, or quarantine centers. Therefore, it is necessary to implement small population breeding programs within these facilities to ensure the correct genetic management of these animals to avoid the loss of diversity.

### 3.5.2. Ex Situ *In Vitro* Organization

Reproductive biotechnologies were developed from 1949 when Polge et al. [2] proposed the use of artificial insemination in cattle breeding. In 1972, cryopreservation of embryos was added to the list of available reproductive biotechnologies [51–53]. Other techniques of cryopreservation involving other types of germplasm (such as tissues, somatic cells, or DNA and isolated genes) are still in their first stages of development [29]. Artificial reproduction methods are well developed and widely employed in breeding programs in all farm animals. These tools are essential for the technological development of breeding programs, but their use in transferring genes from one population to another is lethal for many small local breeds, as a result of the repetitive influences from international breeds.

The use of reproductive technologies can be used for animal biodiversity, including the creation of germplasm banks with the intention to conserve genetic collections of endangered populations. Germplasm banks are long-term stores of genetic materials collected for study, conservation, or breeding purposes. The purposes can differ depending on the intrinsic objectives in open and closed banks.

Open banks permanently receive and produce new germplasm collections. Their objective is mainly the support of breeding programs, but they can also be involved in crossbreeding programs for the creation or maintenance of synthetic breeds. These types of banks can also support conservation programs by increasing the effective population size or participating in the reconstruction of endangered populations. Open banks are routinely used in research projects of different kinds.

Closed banks maintain their collections permanently, and they restrict new accessions to those with genetic significance for the populations. The main purpose of these banks is the support of conservation programs, acting as a reserve to counter problems faced by living populations (inbreeding, indiscriminate crossbreeding, epizootics).

The management of these banks must be organized and coordinated. The information kept in the banks is a collection of data, and it provides descriptions on animal numbers, locations, and the characteristics of animals whose samples are stored [29,54,55]. These inventories must exist at the territorial level, but also at the national and international levels, and all of them must be integrated into a global network. It is essential to register the material location, data bank location, previous movements, container, canister, and color code. The sanitary status of the material and its legal situation (ownership, restrictions) must be recorded as well. The design of the system must consider the following items:

1.  The political organization of the country (territorial, national, and international regulations).
2.  Identification of partners (associations of breeders, private companies, Research and Development centers).

3. Management and coordination (advisory committees, direction committees, scientific committees, and promoters).
4. Financing (governments, NGOs, breeder associations and federations, private stakeholders).

Three types of output must be considered in the budget. The first is the genetic material collection and cryopreservation expenses, which range from the cost of donor animals to the costs of processing equipment and specialist personnel expenses. The second includes the expenses derived from the maintenance of the collection. The third involves the management expenses involving the storage of information and the activities of the committees.

Financial resources are scarce, and it is necessary to prioritize the collections of threatened genetic resources. In this respect, the first decision to be taken is the species, considering the level of endangerment that it faces, availability of specialized methods, technological and human resources, and the present and future value of the species for the country. The second decision to take is the breed to choose within the species, considering the prioritization methods previously mentioned to determine the breed contribution to the general biodiversity of the species [56,57], the genetic situation and value of the breed [58,59], its cultural value [60], or conjoined multicriteria methods [26,27]. The third decision to take is the selection of the recommended individuals within the breed. In this case, eligibility criteria depend on the objectives of the bank. The use and movements of the collections integrated into germplasm banks are affected by the Nagoya Protocol, which regulates how animal genetic resources are utilized, who are the owners, users, and suppliers, and which are the trends in gene flow that these resources describe [12].

New advances enable this process to proceed efficiently in the determination of the best location for the germplasm collections. De Oliveira Silva et al. [61] proposed a mathematical model to optimize logistical decisions of breed conservation and the evaluation of alternative scenarios for efficiently relocating genetic material currently stored in gene banks, allowing for cross-country collections, costs, and cryogenic capacity differentials.

## 4. Management of the International Research

Conservation science involves disciplines including genetics, economics, ecology, assisted reproduction, demography, health, nutrition, sociology, law, food production, and technology, among others. These all contribute to the advances in the characterization, production, improvement, and valuation of the biodiversity for food and agriculture. In all areas, there are important results of investigation, but most of them are isolated activities that sometimes denotes a lack of international coordination. This review focuses on research actions developed by international consortia, considering such initiatives to be the most effective way to achieve the best advances and their maximum dissemination, and favoring the north–south exchange of knowledge.

The most important consortia focus on Europe, given the fact that European Union member countries were already sensitive to conservation, and thanks to the large amount of funding available from the European Union for research on this subject. These consortia incorporate partners from other parts of the world.

In 1996, the work of the EU project European Gene Banking Project for Pig Genetic Resources started. This project involves the four biggest pig-producing countries in Europe: Germany, Spain, France, and Italy. Several publications were developed in the context of the project, and most of the information was compiled in a book by Ollivier et al. [62]. Their efforts and achievements must be acknowledged, as they were pioneers in conducting studies on genetic characterization supported by a consortium [63–66].

This consortium continued as the PIGBIODIV 2 project (Second European pig biodiversity project). New partners from other parts of the world were incorporated into the biodiversity studies on pig breeds from Asia and Europe. This time, the pig industry was involved with the aim of mapping major genes in native breeds with potential interest to the industrial lines [67]. The most important findings

of the genetic characterization and distancing of tens of breeds from both continents were summarized by Megens et al. [68].

Presently, a new consortium for pig research exists, which focuses on the study of demographic parameters, main morphological features, reproductive information, origins and development, and additional information collected at the herd level for European pig breeds [69]. This consortium differs with respect to previous counterparts because it uses a multidisciplinary perspective. The pig consortia proposed a new perspective of pig production where the local breeds reach a new relevance as high-quality producers and as a reservoir of important genes for the industry.

A similar consortium was constituted for the study of cattle genetic resources in Europe. The European Cattle Genetic Diversity Consortium drew important results with regard to cattle evolution [70,71], breed genetic characterization, and distinction [72,73]. They followed a similar trajectory to that of the first pig consortium, and, with their work, the two most important farm species for human feed were covered in terms of their conservation and characterization.

Small ruminants have a specific consortium called ECONOGENE, which integrates molecular genetics, socioeconomics, and geostatistical approaches. These researchers not only consider genetic diversity studies, but also the environment around local breeds of sheep and goats. In this context, Lenstra [74] presented a survey on the evolution and demographic structure of sheep and goats, based on several genetic markers. Simultaneously, other evolutive studies were developed by consortia in defined regions such as North Africa [75], Europe, and the Middle East [76,77]. It is important to especially note a collaboration between ECONOGENE and the European Cattle Genetic Diversity Consortium, focused on the study of the complex spatial pattern of genetic differentiation determining the evolution of the European ruminants, by means of the use of spatial principal component analysis and spatial metric multidimensional scaling by combining geographical information and multivariate analysis [78].

The Equine Genetic Diversity Consortium (EGDC) was formed as an international collaboration team of equine scientific specialists. Its main objective is to quantify nuclear diversity and the relationships within and among horse populations following a genome-wide scale approach by using the Illumina 50K SNP Beadchip.

In birds, an international consortium named AVIANDIV (Development of Strategy and Application of Molecular Tools to Assess Biodiversity in Chicken Genetic Resources), formed in the context of the EU project "Development of Strategy and Application of Molecular Tools to Assess Biodiversity in Chicken Genetic Resources". Its main objective was the study of the chicken biodiversity by means of a custom-designed microsatellite panel, together with the creation of an international database of genotypes [79,80]. An additional German consortium, the SYNBREED (Synergistic Plant and Animal Breeding), developed a new platform based on SNP markers to study chicken biodiversity providing new research insight [81]. This platform is supported by the chicken industry, with the perspective of the use of the local breeds in new perspectives of sustainable production and as a reservoir for important genes linked to the quality of the product and the genetic resistance to illnesses.

Between 2007 and 2011, the GLOBALDIV (A global view of livestock biodiversity and conservation) Consortium [82] was formed. The main objective of the consortium was to gather international experts from different fields related to the characterization of farm animal genetic resources to review the main drivers of biodiversity loss and the main strategies for their conservation. Diversity studies were developed in sheep [76] and goats [83] in collaboration with the GLOBALDIV and ECONOGENE consortia.

Recently, the NEXTGEN (Next generation methods to preserve farm animal biodiversity) consortium was created to go further with regard to the use of new genomic tools to access the landscape genomic topography in several species, combining both molecular and geoclimatic data, to be used in the development of large numbers of models, distinguishing between selection and demographic processes, as described by Stucki et al. [84].

At present, the IMAGE (Innovative Management of Animal Genetic Resources) consortium is active and is devoted to cryobanks of animal germplasm [82] and the use of new genomic tools for their management [85]. IMAGE defines itself as "a consortium to enhance the use of genetic collections and to upgrade animal gene bank management by further developing genomic methodologies, biotechnologies, and bioinformatics for better knowledge and exploitation of animal genetic resources", thereby presenting a nonexclusive interest in conservation but also in commercial breeding.

Most of the consortia mentioned until this point focus on the study of genetic characterization. Some of them enclose a perspective of germplasm banks and in situ conservation programs by including a perspective of the social and/or environmental context of the breeds. Next challenges consist of international cooperation for the valorization of native breeds and their adaptation to the new environmental conditions (climatic change, changes in the use of landscape, etc.) and economic/political situations (Nagoya Protocol and globalization, among others).

Outside the European context, Latin America developed research of local domestic animal genetic resources and their traditional management systems under the umbrella of the CONBIAND network. The CONBIAND network was founded in 1999 [86]. Its main missions are the characterization, conservation, and valorization of Ibero-American local breeds and their traditional husbandry systems, and the evaluation of their ecological and social impacts. The network involves 23 countries from the American continent and other European countries with interest in the region. Some of the most important aims of the network are research, development, innovation, and scientific transference to the sector. These activities are developed by means of internal specific work groups, consortia, and subnetworks devoted to specific subjects. The main approaches or channels implemented by the CONBIAND network concern research, development, innovation, formation, transference, and stakeholder support.

CONBIAND leads several research consortia and working groups whose important activities are described in the paragraphs below, while it also edits and publishes the international indexed journal Actas Iberoamericanas de Conservación Animal (AICA). Innovation is approached through the creation of new tools and methods to improve conservation and valuation actions.

More than 300 Latin American professionals specialized in the conservation of animal genetic resources in specific university training courses. Also, more than 20 PhDs were developed in the context of the network, and a master's degree is currently under creation in this subject. Also, several teaching books were published, mostly focusing on the description of the Latin American domestic animal breeds and their contextualization.

Transference occupies a crucial position for the dissemination to the sector of all the findings and conclusions, promoting and improving farming practices and the quality of life of the farmers. A permanent forum of transference called "Foro Ganadero" (producers forum) is held to stimulate the exchange of information between the sector and the research institutions, so that farmers can immediately benefit from the advances derived from CONBIAND.

Finally, CONBIAND members develop important advising actions for private and public institutions by means of specific reports and their integration in national commissions and committees.

BIOBOVIS (Latin American Bovine Biodiversity Project) was the first consortium to be developed within the network with the aim of doing international research in the scope of native and creole cattle breeds from Ibero-America. The consortium investigated the geo-evolution of creole cattle breeds from America and from Iberian native cattle breeds by using mitochondrial and Y chromosome markers [87–89], and the genetic characterization of such breeds though the use of microsatellites [90]. In the same way, the genetic footprints of the Iberian cattle were traced forward to creole cattle breeds [91], together with the practical evaluation of the conservation priorities of the breeds involved based on molecular information [92]. As a result, the consortium developed a complete analysis of the genetic ancestry of American creole cattle using uniparental and autosomal genetic markers [93].

Simultaneously, the BIOGOAT (Latin American Goat Biodiversity Project) consortium is aimed at the development of biodiversity studies in Ibero-American goat breeds. Its first findings were related

to the genetic relationships between Brazilian and Portuguese goat breeds using microsatellite marker information [94]. Afterward, the consortium studied the population structure of Ibero-American native breeds achieving their primary characterization [95], but also the genetic relationship among Ibero-American goat breeds and their Iberian potential ancestors [96].

The genetic diversity of pig native breeds was studied by the CONBIAND network through its BIOPIG (Latin American Pig Biodiversity Project) consortium. Revidatti et al. [97] developed the genetic characterization of creole pig breeds using microsatellite markers. Later, Cortés et al. [98] studied the genetic relationship among Ibero-American breeds, testing the influence of ancestors, but also establishing the priorities of conservation according to the individual influence of each breed on the total diversity of the species.

Equine biodiversity was also considered within the scope of the CONBIAND network. The fingerprint of Iberian breeds on modern creole American breeds was studied in horses by the BIOHORSE (Latin American Horse Biodiversity Project) consortium [99] and in donkeys by the BIODONKEY (Latin American Donkey Biodiversity Project) consortium [100]. The latter consortium also studied breed contributions to the general biodiversity of the donkey species, outlining some geo-evolutionary conclusions [101].

Three additional consortia recently started implementing their activities: BIOTURKEY (Latin American Turkey Biodiversity Project), BIOVIS, which advanced some preliminary results about the role of the Canary wool-less sheep and all creole breeds of the same type [102], and the BIOCUY (Latin American Guinea Pig Biodiversity Project) consortium, which published research findings on microsatellite tools developed through the studies of biodiversity in guinea pigs, a species with high demand for consumption in the Andean region [103].

A particular subnet of the CONBIAND network is the TRASIBER (Iberoamerican Backyard) network. The TRASIBER network studies traditional management and husbandry systems, especially those linked to family farming, their cultural and social impacts, and their significance for local ecosystems [104]. One of the most recent incorporations to the CONBIAND network is the GASTRIBER (Iberoamerican Gastronomy) working group, dedicated to the rescue of local products and foods as a means to reinforce the conservation of native breeds.

The maximization of the profitability of the outcomes derived from the CONBIAND network provides an excellent worldwide exportable model supported by the many advances that can be achieved through good coordination among research groups and laboratories summing their small resources and conjoining their efforts.

## 5. Conclusions

A national structure to coordinate all territorial actions maximizes the efficiency of use of limited human and economic resources. National structures must be integrated into the FAO to facilitate international cooperation and the exchange of information and experiences.

Non-governmental organizations provide great support for conservation objectives, especially in countries where there is not an efficient governmental organization. These types of organizations flourished in countries with a British cultural context and also in central Europe. One good example of the possibilities of an NGO to implement actions that rely on low collaborative funding in the Latin American is the CONBIAND network.

Recommendations on several aspects of in situ and ex situ conservation are already available, and many countries have their own national programs. In situ conservation national actions present important differences between developed and developing countries. Incentives are the most effective tools for in situ conservation, but methodological transference, technical support, and the creation of protected trademarks for the products also report promising results.

Ex situ in vivo methods are complementary to on-farm conservation, as they maintain live animals, albeit outside their traditional context. It is necessary to have a network of ark, rescue,

and quarantine centers ready to act when facing an emergency at a national level. International rules and recommendations are also needed.

Under extreme situations, the unique methods that guarantee the survival of endangered populations are integrated into ex situ conservation methodologies. Cryopreservation of germplasm ensures the long-term conservation of gene reservoirs in a quantity enough to restore a population to guarantee the minimum level of genetic diversity to maintain its ongoing survival.

National germplasm banks in each country should store the collections of all the breeds of the nation. However, a territorial network of germplasm banks is also recommended to conserve the duplicated collections of the national banks, with the purpose of preventing the loss of resources or collections occurring due to catastrophes.

**Author Contributions:** Conceptualization, J.V.D.B.; formal analysis, A.S., F.J.N.G., J.V.D.B., and M.A.M.M.; funding acquisition, M.E.C.V.; investigation, A.S., F.J.N.G., G.R.G., and J.V.D.B.; methodology, A.S., F.J.N.G., J.V.D.B., and M.A.M.M.; project administration, M.A.M.M. and M.E.C.V.; resources, F.J.N.G., J.V.D.B., and M.E.C.V.; supervision, F.J.N.G. and M.E.C.V.; validation, M.E.C.V.; visualization, A.S. and M.E.C.V.; writing—original draft, J.V.D.B.; writing—review and editing, A.S., F.J.N.G., J.V.D.B., M.A.M.M., and M.E.C.V.

**Funding:** This research received no external funding.

**Acknowledgments:** The authors would like to express their gratitude and thanks to the members of the CONBIAND network and Rare Breeds International.

**Conflicts of Interest:** The authors declare no conflicts of interest.

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
