# Peer review of "Organization and Management of Conservation Programs and Research in Domestic Animal Genetic Resources"

_diversity, doi:10.3390/d11120235_

Round 1

Reviewer 1 Report

My opinion is that the proposed manuscript has been significantly improved.

I have no additional comments to improve the proposed manuscript.

I suggest accepting the manuscript in the proposed form.

Author Response

Thank you very much for your comments.

Reviewer 2 Report

General comments on the abstract:

The abstract should clearly state the objectives of the paper, explain briefly what was investigated and present main results/conclusions. This is completely missing.

It is unclear whether the general statements provided in the abstract are based on a broad literature review (I doubt it), on the findings of the mentioned survey or on recommendations of organisations like FAO.

I disagree with the overgeneralization of several statements like “in situ conservation is a top priority ”as this depends on the context. Indeed currently in situ conservation is the main way how livestock breeds are conserved on global level, but it is considered to be complementary to cryoconservation. For many countries of this world ex-situ in vitro is not "available" as stated by the authors.

Why specifically CONBIAND network is presented is not clear as there are many other organisations active in the field of animal genetic resources.

Some comments on the introduction:

L185: No , the decline became dramatic mid of the 20th century, not in the beginning. Further this statement holds only for Europe.

L186: sensitivity was borrowed???

L188: FAO was not disappointed, but rather concerned

L189: Which ”event” ????

L211:: No, FAO would NOT “update these recommendations in the Second Report on the State of the World´s 211 Animal Genetic Resources for Food and Agriculture later in 2015”. Rather in contrary: The second report was the basis to decide whether The Global Plan of Action for Animal Genetic Resources (FAO, 2007) needs to be updated….

Some comments on Organization of the conservation of Domestic Animals Genetic Resources

L570-577: I disagree with the classification into “agribusiness” versus “Traditional Production”. There are definitions on production systems available (see e.g. http://www.fao.org/livestock-systems/production-systems/en/) that should be used. The authors should check this first. The majority of the 600 million farms in the world are small. Farms of less than 1 hectare account for 70% of all farms…. Therefore smallholders indeed do play an essential role in ensuring food security and nutrition. However, there are a lot of other services (see ecosystemservices) provided by traditional production systems going far beyond food sovereignty, mentioned by the authors.

L552-578: this whole para has nothing to do with the “organization of conservation” and seems rather to express personal feelings of the authors.

L581-583: This statement is based on what? It seems just to express the opinion of the authors.

for the rest of the chapter: What is the purpose of this? The authors express what they think needs to be done regarding the organisation of conservation????

Some comments on chapter 3- International Management:

1062: FAO “assumed” roles????

Chapter 3.2: The overview on NGOs in form of a table showing some key data in a structured way could be helpful. There are much more NGOs, how did the authors choose the ones they try to describe?

L1146 – a special survey on the role of NGO’s was carried out – there needs to be provided more information on the survey! What did the authors really do?

L1212: How do the authors judge that SAVE is the most active NGO in Europe?

Line 1240: This should be section 4, not 3….

L 1252 – 1256: Indeed, there are many attempts to define breed. Is the definition given here the one of the authors????

L1256-1259: What are “original genes”? Are genes lost or rather some alleles? All our genetic patrimony is necessary for human survival and must be preserved? – A more scientific approach to describe the importance of breeds and the risk linked with breed extinction should be considered.

I stop with the detailed comments at this point.

Author Response

General comments on the abstract:

The abstract should clearly state the objectives of the paper, explain briefly what was investigated and present main results/conclusions. This is completely missing.

Response: All suggestions by reviewer have been followed.

It is unclear whether the general statements provided in the abstract are based on a broad literature review (I doubt it), on the findings of the mentioned survey or on recommendations of organisations like FAO.

Response: The abstract has been completely rewritten according to reviewer’s suggestion.

I disagree with the overgeneralization of several statements like “in situ conservation is a top priority ”as this depends on the context. Indeed, currently in situ conservation is the main way how livestock breeds are conserved on global level, but it is considered to be complementary to cryoconservation. For many countries of this world ex-situ in vitro is not "available" as stated by the authors.

Response: We understand the reviewer’s point. However, we think there has been a misunderstanding on some aspects. in situ conservation must be the priority when approaching conservation rather than a complementary action, as it warrantees the establishment and conservation of animals in their specific context. In the same way, we agree ex situ conservation methods are of extreme importance but complementary, but they should apply in cases in which in situ conservation is not ensured. We have tried to delete generalizations as suggested by the reviewer.

Why specifically CONBIAND network is presented is not clear as there are many other organisations active in the field of animal genetic resources.

Response: We mentioned CONBIAND because it has been founded and developed by our team, hence is the example that we know best. We have deleted this specific mention from the abstract, while in the text it has been mentioned as an example of success among this kind of organization, especially in developing countries with low budgets.

Some comments on the introduction:

L185: No , the decline became dramatic mid of the 20th century, not in the beginning. Further this statement holds only for Europe.

Response: We followed reviewer suggestion.

L186: sensitivity was borrowed???

Response: We clarified.

L188: FAO was not disappointed, but rather concerned

Response: we followed reviewer suggestion and changed disappointment by concern.

L189: Which ”event” ????

Response: We apologize for the misunderstanding. “This event“ was a remain of previous edits of the paper. We rearranged the sentence.

L211:: No, FAO would NOT “update these recommendations in the Second Report on the State of the World´s 211 Animal Genetic Resources for Food and Agriculture later in 2015”. Rather in contrary: The second report was the basis to decide whether The Global Plan of Action for Animal Genetic Resources (FAO, 2007) needs to be updated….

 Response: We apologize for the misunderstanding and clarified according to reviewer suggestion.

Some comments on Organization of the conservation of Domestic Animals Genetic Resources

L570-577: I disagree with the classification into “agribusiness” versus “Traditional Production”. There are definitions on production systems available (see e.g. http://www.fao.org/livestock-systems/production-systems/en/) that should be used. The authors should check this first. The majority of the 600 million farms in the world are small. Farms of less than 1 hectare account for 70% of all farms…. Therefore smallholders indeed do play an essential role in ensuring food security and nutrition. However, there are a lot of other services (see ecosystemservices) provided by traditional production systems going far beyond food sovereignty, mentioned by the authors.

Response: The paragraph was rearranged according to reviewer suggestions.

L552-578: this whole para has nothing to do with the “organization of conservation” and seems rather to express personal feelings of the authors.

 Response: We rearranged the paragraph and removed any potential personal appreciation from it as suggested by reviewer.

L581-583: This statement is based on what? It seems just to express the opinion of the authors.

Response: We rewrote the statement to avoid subjectivity as suggested by the reviewer.

for the rest of the chapter: What is the purpose of this? The authors express what they think needs to be done regarding the organisation of conservation????

Response: The present chapter was developed according to the methodologies and schemes present in Spain and in which the authors are directly involved. Spain is one of the countries for which biodiversity reaches the maximum expression in Europe but which has also reached the highest levels of technification and institutional synchronization in the field. However, we checked the section in order to avoid the expressions used to be misunderstood with the mere fact of expressing authors opinions. Anyway, a review, must undoubtfully take into account critical aspects proposed by the authors depending on the state of the art published until the moment. Our experience is supported on 35 years of dedication to conservation and our interaction with authorities and institutions through European projects, FAO agreements and a dense experience involve in the framework of conservation in the countries involved in the CONBIAND network.

Some comments on chapter 3- International Management:

1062: FAO “assumed” roles????

Response: We rewrote the sentence according to reviewer suggestion.

Chapter 3.2: The overview on NGOs in form of a table showing some key data in a structured way could be helpful. There are much more NGOs, how did the authors choose the ones they try to describe?

Response: We choose the present NGOs following international criteria regarding their worldwide status and level of actions. Overview was presented in a Table as suggested by reviewer.

L1146 – a special survey on the role of NGO’s was carried out – there needs to be provided more information on the survey! What did the authors really do?

Response: We apologize for the misunderstanding but the word survey was mistakenly used. This may have been a result s of successive edits of the body text of the manuscript. We corrected it.

L1212: How do the authors judge that SAVE is the most active NGO in Europe?

 Response: We have added information to clarify the comment suggested by the author.

Line 1240: This should be section 4, not 3….

Response: Corrected.

L 1252 – 1256: Indeed, there are many attempts to define breed. Is the definition given here the one of the authors????

Response: Yes, but basing on the common elements of the definitions provided in literature. It was clarified.

L1256-1259: What are “original genes”? Are genes lost or rather some alleles? All our genetic patrimony is necessary for human survival and must be preserved? – A more scientific approach to describe the importance of breeds and the risk linked with breed extinction should be considered.

Response: AN additional paragraph was provided to present a more scientific view of the effects derived from the loss of biodiversity on human health.

I stop with the detailed comments at this point.

Reviewer 3 Report

Dear authors, 

The paper has been substantially improved, but there are still some specific points to be addressed. 

My comments can be found in the attached pdf. 

Author Response

Comments and Suggestions for Authors

Dear authors, 

The paper has been substantially improved, but there are still some specific points to be addressed. 

My comments can be found in the attached pdf. 

 Not sure about this sentence. Did the authors implement a survey by doing interviews with stakeholders? Or did you do an extensive literature review? Please clarify.

Response: As other reviewers suggested the survey section was removed.

livestock can contribute to sustainable development, but is not the only base. There are cases where livestock production does not contribute to sustainable development. do you refer here to specific countries?

Response: We clarified and changed base by one of the main determinants.

Which countries do you mean?

Response: We clarified added ‘developing countries’ to the paragraph.

line 562-576. This section is a justification why conservation is needed, but does not fit under point 2 - organization

Response: We added “why is needed to the title of the Section so that this title fits to the content of the paragraph.

Why do you discuss this difference between agribusiness and traditional production? How is this linked to the managment and organization of conservation? Is there a role of agribusiness in this context? If so, which one. A still have difficulties with the term "traditional production". Is a very vague term.

Response: We understand that it is preceptive to establish the references for each of the macrosystems, as agribussines is in generalr esponsible for the supply of large cities Worldwide, while traditional production is a wode term, not vague, which gathers several production sytems based on previous concepts to production at large scale. This traditional production bears more responsibilities in regards the suppy of small cities and towns and rural livelihoods. Traditional production has an aggregated cultural value which agribussiness lacks.

What we have intended is to make relevant the role of traditional systems in regards conservation of biodiversity of animal genetic resources as agribussiness generally Works with a reduced number of very specialized resources to achieve its aims.

There is a role in agribussiness but this role has a scarce relevancy from the perspective of conservation, as we have already sais, agribussines supports its systems on a reduced number of breeds and strains to produce large quantities of food.

Traditional production gathers together all those systems based upon tradition whose main characteristic is the adaptation to different ecosystems and the large genetic variability considered which are not considered or are far from agribussiness or industrial production.

National Commission of WHAT?

Response: We added the term Genetic resources of agroalimentary use to National Commisions.

Who should be members of the national committee? How are all these stakeholders linked to the committee?

Any proposals how this should work in reality? How do you see this coordination role in operational terms?

Response: We added the following paragraph to respond to the reviewer’s suggestion “The members of National Commissions must be the Ministry of Agriculture, local entities such as local governments, representatives of producers and farmers, federations, universities or research centers scientific representatives and germplasm bank representatives. These structures may function as a network, with local networks which are gathered together under the scope of a National Network, in such a way that, all members are permanently connected.”.

Crossbreeding programs are also selection programs. I assume you talk about pure breeding versus crossbreeding programs. What are biotenology programs. Biotechnology can be an intergral part of the other two programs (e.g. use of AI, embryo transfer etc...)

Response: We implemented changes to fit the reviewer suggestion on which we agree.

Do you mean national level? Optimization: optimal use of available resources.

Response: Clarified.

 This section should be moved up to line 662. It remains unclear who are the members sitting in the committee. Having partnerships with different representatives might only involve their opinion, but might not have the right do make decisions.

Response. We clarified the reviewer comments in the text.

Does the network have a webpage? If yes, would be good to put it there.

Response: RED CONBIAND (http://www.uco.es/conbiand/Bienvenida.html)

This sentence is not clear for me.

Response: Sentence was rewritten.

These are possible strategies, but not actors, who can participate.

I agree all most be involved, but a different level of commitment.

Response: These sections were deleted according to a previos comment by another reviewer.

Does that mean that EU should stop financing conservation efforts? How can the policies be improved?

Response: Of course not. It has been clarified in the text as suggested by the reviewer.

This chapter is basically a review on past international research consortia and their aims. Under the current title I would expect a proposal by the authors how international research could be managed. I propose to change the title of the chapter.

Response: Title was changed.

I would be careful with this statement. The importance varies largely between regions. Do you have data to support this statement?

Response: We agree. The statement was changed and clarified to state that these are two of the most instead of two most.

Yes, CONBIAND has an important role, but there are research projects, which were not executed under the umbrella of CONBIAND. This should be recognized.

Response: Clarified.

do you mean knowledge transfer?

Response: yes, we clarified it in the text.

What does that mean?

Response: Clarified.

This manuscript is a resubmission of an earlier submission. The following is a list of the peer review reports and author responses from that submission.

Round 1

Reviewer 1 Report

The proposed manuscript is too general. Very relevant references were not used in the preparation of the manuscript. For example: FAO (2012): Cryoconservation of animal genetic resources.

The proposed manuscript ''Organization of the conservation of Domestic Animal Genetic Resources. A review'' is not a scientific paper.

The proposed manuscript does not bring new ideas or strategies. My opinion: it is too general and not innovative.  

Very relevant references were not used in the preparation of the manuscript. For example: FAO (2012): Cryoconservation of animal genetic resources. http://www.fao.org/3/i3017e/i3017e00.pdf

Reviewer 2 Report

The level of English is extremely poor, the terminology used not adequate. Several paragraphs express a very subjective opinion of the authors (e.g. on the role of agrobusiness).

The paper needs a thorough language editing, further the terminology used is not the one used in the respective literature (quite strange for a review paper that should be based on a sound literature review). It neither provides a comprehensive overview, nor any vision, but contains some quite subjective statements as mentioned in my short review.

Reviewer 3 Report

The manuscript entitled :

"Organization of the conservation of Domestic Animal Genetic Resources. A review" is a review explaining the importance of National Germplasm Banks. Authors described the main steps for the establishment of conservation consortia, such as CONBIAND. 

Although I do not fully recognize the merit of such review, I do not see issues with the paper. It is fluent, describing all aspects of the problem and sending a clear message. To increase the quality of the text, I strongly recommend a review from the native speaker or check the manuscript with the Grammarly. I run a few paragraphs trough it because some of the sentence constructions sounded a bit off, and there were substantial improvements. 

Furthermore, I would remove "A review." from the title. 

Reviewer 4 Report

Dear authors, 

I have read your manuscript with great interest. Although I came to the decision to reject the manuscript, I really encourage you to improve it for a new submission. I think that the topic is of high relevance for the animal breeding community. 

By addressing the topic of organizational questions of conservation of domestic animal genetic resources the authors address an interesting topic in their manuscript. But in my opinion the title only partly fits the body of the text. In general, the text is too long and there are many redundancies, which have to be avoided. I find it interesting to see the evolution of FAO´s engagement in the topic and helps the reader to learn about the history of the different steps taken by the international community.
The manuscript is a mixture of reviewing the history of conservation strategies and a manual of how conservation strategies should be organized.
The introduction is far too long as domestication processes and related topics such as genetic drift, genetic erosion etc has been described in other documents in detail. It is enough to refer to some key publications.
P2L61. What is “Illustration period”?
P2L65-66. I do not agree that the use of industrial agricultural by-products is called “Application of artificial feeding”. This is a well-documented way of improving the feeding base in many livestock production systems by exploiting all available resources. I agree that here might be extreme cases, where limits are reached.

I do not know what is meant by “animal feeding specific agriculture”. I assume the authors mean “concentrate feeds”.
P2L68ff. I agree that migration might be one cause of loss of diversity, but is definitely not the only one. Depending on the world region, different causes can be identified.
Coordination of the conservation
This chapter describes in a very detailed way how coordination at national level should be organized. The authors propose a National Program for the Conservation of the Agriculture Biodiversity, ignoring already existing mechanisms, policies and agencies in different countries. I am also not confident with the detailed structure and mechanisms outlined. E.g. the authors propose annual meetings and agenda of such a meeting. E.g. breed recognition is described P6 L205-221. This could be shortened.

Figure 1. I do not agree with the authors that agribusiness is responsible for “food safety”. This term refers to ensure that people have access to “safe food”. This should be also ensured by “traditional production”. Maybe the authors mean “food security”? I have also difficulties with the argument that traditional production is responsible for food sovereignty. First, the term “traditional production” is a very vague concept and the concept of “food sovereignty” has be discussed/explained in this context. I also do not see this clear dichotomy of agribusiness versus traditional production. These two systems indicate the extreme points, but along this gradient there is a huge diversity of production systems, all of them hopefully contributing to production of safe food.

P7L259-267. The authors propose a network of germplasm banks at national scale. This is a good idea, but might not be feasible/necessary for very small countries.
This chapter requires restructuring. There is this long description of networks of germplasm banks, which is followed by a general description of activities done by FAO. Information which has been provided partly at the beginning of the manuscript.
P8L310-312. The authors mention that a special survey on the role of NGOs was carried out in the present review. How was this review performed? How was data collection done? What was the purpose of this survey? What are the main findings?
Organization of in situ conservation
P9L383-385. The authors provide a breed definition, but without a reference.
Organization of ex situ conservation
P11.L485-486. “Centers are identified keeping the population context as close as possible”. What does that mean?
P11L495-502. I cannot follow the argument of the authors of the implementation of breeding programs for small populations. I agree that these are important breeding programs, but in case of a major disease outbreak they might not be very helpful in avoiding the loss of genetic diversity.

P11L505-506. What is meant by “loss of single populations was noticed to affect the equilibrium of global ecosystems”? This might be the case for wildlife, but I do not think that the global ecosystem is affected by the loss of a single breed.
Organization of cooperation in international research
P13L610. What is “General directorates VI and XII”?
This chapter is a long description of previous research networks/projects. This could be done in a more in-depth analysis of these projects. What are challenges? What are benefits? What are commonalities and in which aspects are they different?
CONBIAND is highlighted as a good example, but the authors do not provide insights what are the “ingredients” of success. This would be extremely interesting to understand and learn from this initiative.
Conclusions

The conclusions are far too long and repetitive. The authors argue that a financial support from North to South has to be intensified. Is this really the only solution? How do you argue that countries from the North should pay for the conservation of local breeds in the South? What is needed from the countries in the South? Some might be actually in the economic position to finance conservation strategies, but the political will/awareness is not given. The authors do not mention that one of the targets and two of the indicators of The UN Sustainable Development Goals is related to conservation of genetic resources.
Target: 2.5. By 2020, maintain the genetic diversity of seeds, cultivated plants and farmed and domesticated animals and their related wild species, including through soundly managed and diversified seed and plant banks at the national, regional and international levels, and promote

access to and fair and equitable sharing of benefits arising from the utilization of genetic resources and associated traditional knowledge, as internationally agreed.
Indicator 2.5.1: Number of plant and animal genetic resources for food and agriculture secured in either medium or long-term conservation facilities
Indicator 2.5.2: Proportion of local breeds classified as being at risk, not-at-risk or at unknown level of risk of extinction
Further information: https://medium.com/sdgs-resources/sdg-2-indicators-3a59a1c210b0
The authors use sometimes incorrect/not precise nomenclature.
• In the title authors use the term “domestic animal genetic resources”, but later on in the text always use the word “zoogenetic”. This term is used in Spanish “recursos zoogenéticos”, but FAO uses the term AnGR-animal genetic resources.
• P1 L44 “new genetic forms”: what does that mean?
• P5L11-179. The authors sometimes use the term “regional programs”, but later on they refer to “territorial programs”. Is this the same meaning?
• P5L186-187: “…comprise of different partnerships”. Do you mean “stakeholders”?
• P11L463 “anarchic crossbreeding” – I assume the authors refer to “indiscriminate crossbreeding”.

The text requires substantial language improvement.
While reading the manuscript I thought that the manuscript could be re-organized. I propose that the authors re-think the structure of the manuscript. Maybe it would make more sense to put the CONBIAND network in the center and explain this case study (benefits, challenges, network structure, etc…) and embed it in the broader context of organization of conservation of animal genetic resources.
At the current state, I have to reject the paper, but encourage the authors to improve the manuscript for a new submission.